# Paediatric Cutaneous Warts and Verrucae: An Update

**DOI:** 10.3390/ijerph192416400

**Published:** 2022-12-07

**Authors:** Ivan Bristow

**Affiliations:** Private Practice, Lymington, Hampshire SO41 9AH, UK; mail@ibristow.com

**Keywords:** pediatric warts verrucae prevalence HPV

## Abstract

Cutaneous warts are common lesions in children caused by the Human Papilloma Virus (HPV) and for most lesions spontaneously resolve within months of the initial infection, regardless of treatment. The infection is most prevalent in the second decade of life affecting over 40% of children. Studies have demonstrated wart virus carriage on normal skin is higher in children with active lesions and family members. Subtypes HPV 2, HPV 27, HPV 57 and HPV 63 are particularly common in paediatric populations. Warts arising on the plantar surface of the foot (verrucae) can be particularly problematic owing to the location. They may interfere with daily activities causing pain and embarrassment. Plantar lesions have been shown to be more resistant to treatment than warts elsewhere on the skin. Systematic reviews and studies conducted over the last decade have demonstrated little evidence of innovation or effective improvements in treatment of recalcitrant lesions over the last 30 years. However, newer modalities such as immunotherapy (using injected vaccines) and hyperthermia using microwave treatment may hold promise in improving the treatment of these common and therapeutically frustrating lesions.

## 1. Introduction

Warts and verrucae are benign growths occurring in the skin and mucous membranes due to infection with various strains of the Human Papilloma Virus (HPV). Depending on their location and morphology they may be referred to as common, plane, flat, plantar or genital warts. This paper aims to review recent evidence from published reviews and other research regarding cutaneous (non-genital) warts in children with a specific focus, where possible, on plantar lesions (verrucae).

### Prevalence and Risk Factors for Warts

The cutaneous manifestations of HPV infection are cutaneous warts and verrucae, which have been shown to be unusual before the age of 4 [1] but most prevalent in secondary school aged children. A cross-sectional study of the hands and feet of 1465 children aged between 4 and 15 from four Dutch schools showed a prevalence of 33% with most children having only one or two lesions with no gender predilection. Prevalence rates within this study demonstrated a 15% prevalence in four-year-old school children rising significantly to 44% in eleven-year-olds. Within this cohort, 59% of children exclusively had plantar lesions, with 13% showing hand and foot involvement [2]. Warts were more common in Caucasian than non-Caucasian children. Interestingly, barefoot activities and swimming pool visits did not increase the risk of developing warts. In other work conducted on children with warts [3,4,5] additional risk factors identified included Caucasian skin type, sharing shoes, coming from a large family, having a father who was a manual worker and having siblings or classmates with warts. Regional geographical variations in incidence were also noted in one UK study of schoolchildren [5]. Data on prevalence rates in older children and young adults are scarce but one study of over 15 000 Chinese college students suggested a prevalence of around 1.4% [6]. The presence of verrucae in a group of children with rheumatological disease showed a prevalence no higher than the prevalence in healthy children, even though most were known to be taking immune modifying medication. Lesions in these patients were no more numerous or atypical compared to other children [7]. 

## 2. HPV Type and Carriage of Infection

Cutaneous warts are caused by small DNA viruses which have adapted to infect the skin and mucosal surfaces. Over 450 species have been identified amongst the five genera of the HPV viruses (alpha, beta, gamma, mu and nu) [8], based on genotyping of the L1 capsid gene. The HPV virus responsible for cutaneous warts can be routinely detected from normal skin swabs in children. However, skin swabs of wart free skin are more frequently positive in children who have warts elsewhere. The sole of the foot has been found to show highest positivity for wart virus carriage on normal skin in a paediatric population [9].

Few studies have investigated the most common HPV sub-types responsible for causing cutaneous warts. Researchers in the Netherlands selected 31 children (aged 10–12 years) from three schools who were medically diagnosed with warts on the hands and feet. Skin swabs taken from their lesions and samples were analysed for the presence of HPV DNA using a polymerase chain reaction (PCR) test assay capable of being detected in 23 known HPV types. In total HPV was detected in 92% of warts. In 40 plantar lesions, the most common sub-types were HPV 2, HPV 27, HPV 57 and HPV 63 (but HPV 1, HPV 4, HPV 10, HPV 41, HPV 65, HPV 88 and HPV 95 were also detected). A small number demonstrated multiple HPV strains within the same lesion [10]. These findings broadly concur with a Spanish study which exclusively genotyped plantar warts (n = 105) which were all positive for one HPV subtype. The most prevalent genotype was HPV 57 (37.1%), followed by HPV 27 (23.8%), HPV 1a (20.9%), HPV 2 (15.2%) and HPV 65 (2.8%). In children under 11 years of age HPV 1a was the most prevalent type [11]. Tomson and colleagues in their study reported HPV types 2, 27 and 57 as being the most common cause of plantar lesions [12].

## 3. Diagnosis

The diagnosis of warts is predominantly clinical, visual analysis of the lesion being the most common method, particularly as the cost of genotyping using polymerase chain reaction (PCR) precludes its use in regular day-to-day practice in many countries. In children, there are few differential diagnoses that may mimic plantar warts. A study undertook development of a visual assessment tool to assist clinicians in the identification of cutaneous warts. The CWARTS tool [13] was developed to test inter-observer agreement on nine clinical features of warts. Good inter and intra-observer agreement was achieved with the presence of black dots within the lesion having the highest agreement amongst observers [13] and a latter publication showed it to be the strongest predictor of HPV presence within a lesion [14].

In recent years, the use of dermoscopy has increased in the assessment and recognition of skin lesions. Dermatoscopic evaluation of the visual features of warts has been undertaken. Observable reported features in warts include the presence of dots and globules—these can be red, brown or black in colour and probably represent dilated capillaries [15]. These capillaries are situated at the centre of white halos [16] giving a frog spawn like appearance (Figure 1). In some lesions, abrupt interruption of the natural dermatoglyphics is easily observable under the polarised light of the dermatoscope. Callus, a common differential diagnosis on the sole, lacks the typical dots and globules seen in warts whilst there is no interruption in the natural dermatoglyphics (Figure 2). In addition, callus displays central reddish to bluish structureless pigmentation [17].

## 4. Natural Wart Regression

It is well established that cutaneous warts and verrucae may undergo spontaneous regression—the underlying mechanisms responsible for this are not fully understood however several observational studies of warts in children have reported spontaneous clearance. A follow up study of 364 children aged eleven with warts discovered only 7% had them by the age of sixteen—demonstrating a 93% regression rate [5]. A Dutch study of 366 schoolchildren diagnosed with warts reported 50% of subjects were wart free within a year. In addition, a younger age and non-Caucasian ethnicity was shown to favour faster resolution [18]. In a further study by the authors, it was demonstrated that the HPV sub-type of plantar warts influenced outcomes following treatment, particularly for children with the HPV1 type. HPV1 induced plantar lesions were more likely to regress with no treatment versus HPV 2, HPV 27, HPV 57 (58% versus 7%). Moreover, following treatment with salicylic acid the HPV1 type was more likely to respond to treatment than HPV2, HPV 27 and HPV 57 types (92% versus 25%) [19]. This work suggests that wart typing may allow optimisation of treatment as identification of the plantar warts caused by HPV 1 type have an eightfold chance of spontaneous regression. Other HPV types causing plantar lesions may confer more resistance to treatment, but this warrants further investigation.

## 5. Treatment of Warts in Children

In the last ten years there have been several systematic reviews undertaken investigating wart treatments and outcomes. Some have focused exclusively on plantar lesions [20,21,22]; others have included all types of cutaneous warts. Other reviews have pooled adults and paediatric data [23,24] in their analyses or have looked exclusively at paediatric data [25,26]. These reviews have uncovered areas of uncertainty in many aspects of wart treatment and have renewed calls for more research to investigate suitable treatment regimens using robust methodologies. 

In 2014, a published systematic review [24] examined the effectiveness of various cutaneous wart treatments in adults and children concluding that topical salicylic acid showed benefit with cryotherapy and immunotherapy showing less convincing evidence of effectiveness. Other therapies such as bleomycin, photodynamic therapy, duct tape, surgical excisions, lasers and candida antigen had unknown effectiveness. Similar guidelines published in the same year from the British Association of Dermatologists [23] echoed the findings, concluding that most evidence of effectiveness for salicylic acid over cryotherapy with other therapies needing more investigation to draw conclusions. Within this paper it was also reported that plantar warts were likely to show reduced levels of responses compared to cutaneous warts elsewhere on the skin, reporting response rates to salicylic acid and cryotherapy of just over 30% for plantar lesions.

Systematic reviews of the treatment of warts in children specifically are lacking. Warts in children are known to regress more quickly than in adults, which may give different efficacy rates to adults. In addition, younger patients may be less tolerant of painful interventions limiting the available treatment options. In 2020, a review of topical treatments for cutaneous warts in children was published [26], reviewing 38 published papers from 1969–2021. Papers were only included if they held specific paediatric data but excluded case series and case reports. A wide range of interventions were included and detailed, but no systematic analysis was performed. The authors conclude with a suggested algorithm for children and adolescents suggesting plantar lesions in younger or uncooperative children should be approached with a “watch and wait” approach or salicylic acid-based preparations. Older children may be treated with a similar approach, but cryotherapy could also be employed as an alternative. Stubborn lesions failing these modalities may be considered for third line treatments such as immunotherapies or laser treatment where less evidence of effectiveness exists.

## 6. Treatment of Plantar Warts (Verrucae)

A recent systematic review focused exclusively on studies reporting treatments for plantar warts. The authors only included studies which had 100 or more subjects [21]. A total of nine studies involving 1557 patients (adults and children) met their inclusion criteria. The review found most research published around salicylic acid (in various concentrations and in some studies compounded with other agents), cryotherapy (with a small number reporting the use of carbon dioxide) and pulses dye lasers. They concluded that salicylic acid showed broadly similar outcomes to cryotherapy but was more cost-effective. In addition, there was modest evidence that lasers may demonstrate positive benefit but evidence for many other modalities was lacking (curettage, cautery, photodynamic therapy, intralesional injections and zinc therapy).

A systematic review published in 2022 [21] reviewed randomized controlled trials involving cryotherapy of plantar warts (including patients of all ages). A total of 14 papers met the inclusion criteria and were evaluated (1084 patients, average patient age was 27 years). The review included comparisons of differing cryotherapy regimes (method of application, timings, etc.) and trials comparing cryotherapy with other treatments (40% trichloroacetic acid, duct tape, carbon dioxide laser, 10% formaldehyde, adapalene gel, radio frequency ablation, salicylic acid, Nd:YAG laser and acyclovir). The work concluded that there was no evidence to suggest that cryotherapy was either superior or inferior. 

Another study, published earlier in 2020, systematically reviewed topical treatments for plantar warts including patients of all ages [20]. The review showed that traditional first line and second line treatments such as salicylic acid and cryotherapy showed lower rates of effectiveness for plantar lesions concurring with the earlier review by Sterling et al., [23]. The authors suggest that an alternative approach may be required, such as immunotherapies with intralesional treatments. However, firm conclusions could not be drawn, as despite more impressive cure rates, the data was drawn from lower-level studies such as case series, prompting the call for more robust research into these treatments. 

As the above reviews have highlighted, intralesional treatment for paediatric warts may have a place as a third line treatment for more resistant lesions, particularly for the less responsive plantar warts. For natural clearance of warts, it is generally accepted that cell mediated immunity is required. Immunity is facilitated through both innate and adaptive pathways within the skin. Intralesional immunotherapies increase host cell ability to recognise and eradicate HPV infection by facilitating cell mediated immunity. One review qualitatively evaluated studies which included the use of Candida antigen, the Mumps-Measles-Rubella vaccine (MMR), Tuberculin or Bacillus Calmette-Guerin (BCG) as a an intralesional injection. A total of 20 studies were identified (which included adult data). Complete resolution rates ranged from 39–88% for candida, 26–92% for MMR, 23.3–94.4% PPD and 33.3–39.7% for BCG. One study did show higher clearance rates in its younger patients (MMR and PPD). Local side effects for these treatments include injection pain or burning sensation or blistering whilst use of these agents have caused oedema, fever and myalgia in some cases. 

The HPV vaccine has been available for several years to protect against oncogenic strains of the HPV responsible for the majority of genital and anal cancers. Case reports have been published reporting cutaneous wart clearance following vaccination with the quadrivalent vaccine (Gardasil^®^) which is active against HPV strains 6, 11, 16 and 18 [27,28]. Following this, a study of 6 children with warts persisting for more than 2 years (which had failed other therapies) was undertaken [29]. They received three intramuscular injections of the vaccine at 0, 2 and 6 months. At 8 months all lesions had regressed. Whilst a promising result, the authors highlighted how the response was probably age related, as older children and adults who had received the same treatment latterly showed disappointing response rates. They postulated that the response was more apparent in pre-pubescent children. With the beginning of puberty, the major histocompatibility complex (MHC) class I molecule of the HPV-infected cells disappears, resulting in a decrease in the vaccine-induced HPV-specific cytotoxic T-cell immune response. This could have been a factor explaining the age-dependent therapeutic response to the vaccine [29].

## 7. Emerging Treatments

The HPV vaccine has been available for several years to protect against oncogenic strains of the HPV responsible for the majority of genital and anal cancers. Although previous authors have remarked on the scant progress in efficacy for wart treatments in the last few decades, with a lack of new treatments [21,30], new therapeutic approaches are being developed clinically. Research in recent decades has focused on the effects of heating tissue to around 41–44° degrees centigrade (termed the hyperthermic range) which is non-lethal to cells but has been shown to promote immune function within tissues [31]. Previous work has established that wart persistence exists due to mechanisms induced by HPV infection that down regulate antigen processing and presentation, suppressing the normal immune response [32,33,34]. Heating tissues to within the hyperthermic range has been shown to elicit the release of Heat Shock Proteins (HSP) from cells which have several immune promoting effects, including maturation and migration of Langerhans cells, particularly in HPV infected skin [35], increasing cytokine and interferon release [31,36]. Clinically, increased clearance of plantar warts has been observed in a randomised controlled trial where 54% of the hyperthermia treated group resolved, as opposed to just 12% in the placebo arm [37]. Similarly, a study of 29 warts treated with controlled heating or placebo favoured the former (86% resolution versus 41% of untreated lesions) [38].

Clinically, the use of hyperthermia by way of a microwave device has shown benefit in the treatment of plantar warts. In the laboratory a study of the effects of the microwave device on skin explants showed increased antigen presenting cell activity and presentation to CD8+ T cells and increased γ-interferon release [39]. A clinical study undertaken by the author [40] treated 54 treatment resistant warts in 32 adults. Following four treatments, a clearance of 75.1% (41 warts) clearance of lesions was noted. Further clinical research is currently underway.

## 8. Summary

Cutaneous warts affect a significant number of children with limited evidence of effective treatment in this patient group. The most recent systematic reviews report little change in the last 20 years in the evidence base, with topical treatments such as salicylic acid and cryotherapy being the most studied. These show moderate effectiveness in paediatric and adult populations. Newer treatment modalities such as injection or immunotherapies and microwave heating show promise but further robust research is required to ensure effectiveness in this population group.

## Figures and Tables

**Figure 1 ijerph-19-16400-f001:**
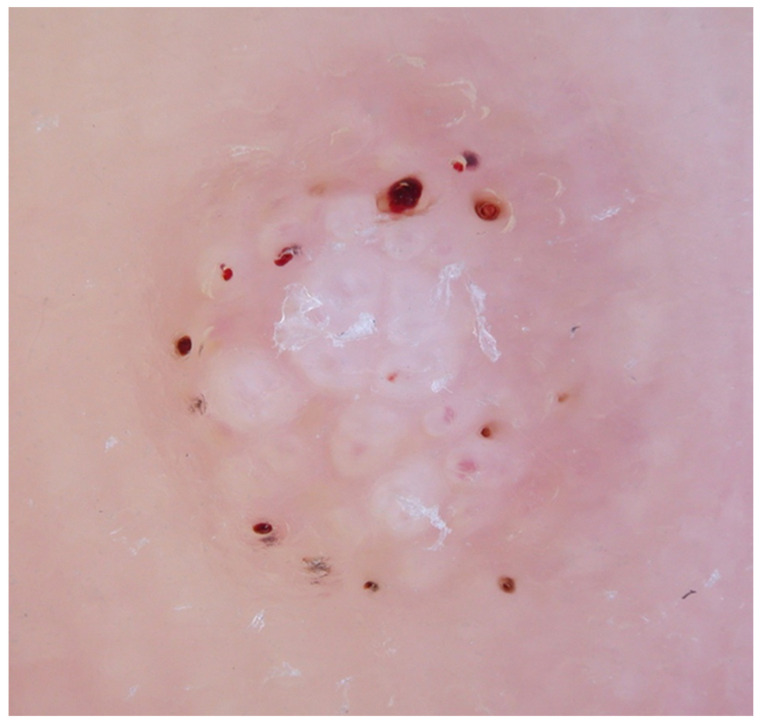
Dermoscopy of a debrided plantar wart showing a frog spawn like appearance with white halos, with capillaries at the centre of the lesion.

**Figure 2 ijerph-19-16400-f002:**
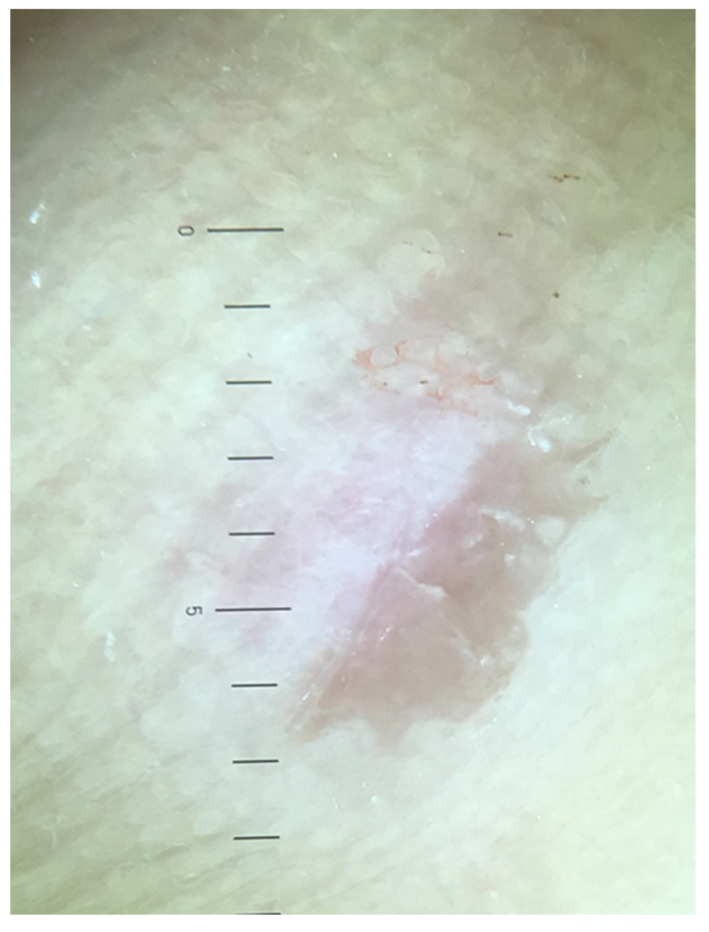
Dermoscopy of callus showing a central featureless reddish area.

## Data Availability

Not applicable.

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
