# Peer review of "Paediatric Cutaneous Warts and Verrucae: An Update"

_ijerph, 2022, doi:10.3390/ijerph192416400_

Round 1
Reviewer 1 Report
Thank you for submitting a relevant, timely and interesting article providing a concise and highly informative update on paediatric cutaneous warts and verrucae. This will be of interest to a wide range of audiences, most notably GPs, dermatologists and podiatrists. I have no amendments to suggest and would commend the author for the quality of writing and its concision. The remarks on emerging treatments is particularly helpful, given the rather moribund range of treatments to date.
This paper provides a detailed update on the various types of cutaneous non-genital warts and plantar verrucae found in children, including information on the viral subtypes of the HPV virus most commonly found to infect children; prevalence and risk factors; and systematic review data on the effectiveness (or otherwise) of common treatment modalities. Interestingly, it adds further information on the more recent data on newer modalities of treatment, most notably immunotherapy and hyperthermia using microwave treatment. In addition, it includes a useful current guide to clinical diagnosis, including visual assessment tools (such as CWARTS) and the use of dermoscopy. Given that misdiagnosis remains a common problem for a condition largely dependent on clinical recognition, it is most helpful to have this section included, particularly with dermatoscope images which enhance accuracy in diagnosis.
The key strengths of the article lie in its clarity and comprehensive coverage of current knowledge on the types and treatments of paediatric cutaneous warts and verrucae, alongside its handling of the contemporary use of immunotherapy and microwave technology in their management, given the latter provide much more encouraging signs of success than has been the case for most standard treatments (from salicylic acid, cryotherapy, lasers and surgery to the use of occlusive duct tape). The latter part of the paper focuses particularly upon the treatment of plantar warts (verrucae), which is a logical step in light of the data indicating that plantar warts tend to be more resistant to treatment (a point addressed in the section dealing with HPV types in cutaneous wart infections). The data sources (such as the range of systematic reviews) are contemporary and thus based on the most recent available material, and each are presented in a coherent and concise way, as a useful summary guide with clear clinical implications. The section on immunotherapy approaches provides clear information on the use of vaccines of various types and their relative success rates, again with implications for clinical practice.
Perhaps of the greatest relevance is the section dealing with emerging treatments, which, although short, provides early indications of the effectiveness of newer treatments (microwave hyperthermia is clearly the most promising avenue) and the likely mechanisms of action. I note that the author declares an advisory role with the company Emblation, but without serving any role in the production of the paper.
I have checked and re-checked the article for grammatical or typographical errors, and, unusually, I have found none (which is to be commended). The article is brief – it is intended to be so, as an update – and yet its brevity is a strength, given the clarity and succinctness of the writing. Nor is it particularly easy to identify major recommendations for improvement, as the paper’s strengths are not matched by a similar range of weaknesses. The majority of the papers reviewed are post-2012 (within the last decade) although the earliest cited is 1998. This is not problematic, considering that the treatment of plantar warts has remained largely unchanged over quite a long period – until very recently, which is, in part, the reason and purpose of the paper.
Of course, the article serves as a short update, rather than an extensive review of the literature, but this carries with it the advantage that it provides an excellent summary of recent advances and their relevance to clinical practice, which is likely to be of considerable value to the audience (assuming the audience to primarily include GPs, dermatologists and podiatrists).
In light of the quality of the writing, the informative and succinct content, the helpful sections neatly sequenced through the paper, and the absence of typographical or grammatical errors, I am satisfied that the paper is suitable for publication without further amendment.
Author Response
Thank you for your comments.
Reviewer 2 Report
The terms "cutaneous warts" and "verrucae," to me are interchangeable. However, within the paper, there seems to be the word "and" used often between the terms or they are used separately in other locations in the paper.
Is there a difference?
Author Response
Thank you for your review and comments. I have amended the first paragraph to discriminate between cutaneous warts and plantar warts (verrucae). This is to enable the paper to highlight where study data is focussed particularly on plantar lesions as opposed to cutaneous warts elsewhere which I feel is more informative for the reader.
Reviewer 3 Report
Thank you for the opportunity to review “Paediatric cutaneous warts and verrucae: an update”. The manuscript is well-developed and a pleasure to read. Furthermore, it covers a very common concern, particularly for those working with children, and I have no doubt practitioners will appreciate the succinct and current nature of this article and the evidence it summarises.
I have a few minor editorial suggestions below but would otherwise be supportive of this being published.
Line 75 change “way in which warts are diagnosed” to “method”.
Line 93 - 95 requires attention as some formatting suggestions appear to be incorporated, and the final sentence appears out of place.
Lines 102 – 103 the first two sentences could be combined.
Line 114 – remove ‘prior to treatment’ or change the second half to not repeat ‘treatment’.
Lines 127 – 134 appear to contradict themselves regarding whether salicylic acid is preferred. I suspect a wording change is required either in the first sentence or after “echoed the findings”?
Line 208 is a repeat of a line introducing the previous paragraph. A reword would improve readability.
Author Response
Thank you for your comments. I have made amendments in the text as suggested to improve readability.
Reviewer 4 Report
Thank you for this helpful contribution. Thank you for sending your paper entitled “Paediatric cutaneous warts and verrucae: an update” to IJERPH. After carefully review this interesting paper, the following comments are listed for your reference:
The authors present a review of Paediatric cutaneous warts and verrucae: an update. The subject is very interesting, besides the results are important. This paper aims to review recent evidence from published reviews and other research regarding cutaneous warts in children. The author present the prevalence and risk factors for warts, and HPV type and carriage of infection. In the last years there have been several systematic reviews undertaken investigating wart treatments and outcomes. I write some comments below that could benefit the article:
-It would be interesting if the author presented a table with the most relevant results and variables of the studies analyzed.
-Have you registered the revision in some registry base?
-I recommend reviewing the instructions for the author. We recommend preparing the references with a bibliography software package, such as EndNote or Zotero to avoid typing mistakes and duplicated references. Please check the instructions for authors for the relevant journal and article type for examples of the relevant reference style.
The authors could improve their manuscript based on the reviewers' recommendations. Thank you for this invitation to provide a review to evaluate this article for publication. I am delighted to have been invited to review this work.
.
Author Response
Thank you for the comments. The paper was formatted using Endnote software. I have not included the suggestion of a table, purely as to do this thoroughly would require the paper to be re-written, taking on the form of a systematic review, when its focus is on current and emerging therapies.
Round 2
Reviewer 4 Report
Thank you for this invitation to provide a review to evaluate this article for publication. I am delighted to have been invited to review this work.